# The Elaboration of an Intersectoral Partnership to Perform Health Impact Assessment in Urban Planning: The Experience of Quebec City (Canada)

**DOI:** 10.3390/ijerph17207556

**Published:** 2020-10-17

**Authors:** Stéphanie Gamache, Thierno Amadou Diallo, Ketan Shankardass, Alexandre Lebel

**Affiliations:** 1Graduate School of Land Management and Regional Planning, Faculty of Planning, Architecture, Art and Design, Laval University, Québec, QC G1V 0A6, Canada; stephanie.gamache@crad.ulaval.ca (S.G.); thiernoamadou.diallo@inspq.qc.ca (T.A.D.); 2National Collaborating Centre for Healthy Public Policy, Montréal, QC H2P 1E2, Canada; 3Department of Health Sciences, Faculty of Science, Wilfrid Laurier University, Waterloo, ON N2L 3C5, Canada; kshankardass@wlu.ca; 4MAP Centre for Urban Health Solutions, St. Michael’s Hospital, Toronto, ON M5B 1W8, Canada; 5Quebec Heart and Lung Institute, Québec, QC G1V 4G5, Canada

**Keywords:** health impact assessment, urban planning, public health, knowledge translation

## Abstract

Health impact assessments (HIA) allow evaluation of urban interventions’ potential effects on health and facilitate decision-making in the urban planning process. However, few municipalities have implemented this method in Canada. This paper presents the approach developed with partners, the process, and the outcomes of HIA implementation after seven years of interinstitutional collaborations in Quebec City (ten HIA). Using direct observation and meeting minutes, information includes: perceived role of each institution taking part in HIA beforehand, how the HIA process was implemented, if it was appreciated, and which outcomes were observed. The intersectoral interactions contributed to the development of a common language, which sped up the HIA process over time and fostered positive collaborations in unrelated projects. It was an effective tool to share concerns and responsibilities among independent institutions. This experience resulted in the creation of an informal group of stakeholders from four different institutions that perform HIA to this day in collaboration with researchers.

## 1. Introduction

### 1.1. Historical Background: The Emergence of Health Impact Assessment

In the early 1970s, positive aspects of economic development were identified as having adverse impacts [1]. The next decades saw high-income countries enjoying a period of both peace and economic expansion, while many other countries were left in poverty [2]. The economic growth spread with the common aim for people and Governments across this expanding global system of tackling ill health, poverty, and other related societal problems [3].

People, from countries of all income levels benefited from advances in sanitation, nutrition and vaccination [4]. However, this great acceleration was accompanied by the rise of many non-communicable diseases, injuries and growing rates of mental health issues, increasing demand on health systems [5]. Today, as over 50% of the population live in cities, with 80% in high-income countries, an increasing number of studies report adverse effects of the built [6] and social environment [7] on health and health determinants, either through direct or indirect pathways [8].

During this period, these concerns motivated several initiatives to tackle the consequences of this unprecedented economic growth. These initiatives led the World Health Organization to propose the Ottawa Charter for Health Promotion (WHO, 1986), which contributed to putting health promotion on the international agenda [9,10]. Many countries responded positively to the Ottawa Charter and a great variety of policies were produced with the aim of creating social and built environments supporting health and positive health behaviors [9].

This charter recognized that decisions in non-health related sectors often have an impact on health and stressed the necessity to build intersectoral approaches to create supportive environments. In this context, it is necessary to develop tools to promote and strengthen intersectoral collaboration between the health sector and other sectors and to integrate health issues into all policies. One way to achieve this is through the implementation of Health Impact Assessment (HIA) in the decision-making and urban planning process [11].

### 1.2. What Is an Health Impact Assessment (HIA)?

HIA is a flexible tool supporting decision making that enables the consideration of health in the development of policies, programs, or projects, particularly outside the health sector [12]. It was developed during the 1990s to complement environmental impact assessments and to better take into consideration social and health inequalities [13]. HIA has been defined by the WHO as “a combination of procedures, methods and tools by which a policy, program or project may be judged as to its potential effects on the health of a population, and the distribution of those effects within the population” [14], and “identifies appropriate actions in order to manage those effects” [15]. The HIA procedure consists of five steps [16,17], as shown in Figure 1.

HIA has been used in many developed countries, such as the USA [18], New-Zealand, Australia [19], some Asian and African countries [20,21] and many European countries [22].

### 1.3. HIA in Canada

The development of impact assessments in Canada dates to the early 1970s, with the formal introduction in 1973 of the Environmental Impact Assessment (EIA) at the federal level. In 1992, the Canadian Environmental Assessment Act was introduced and by 1995 it began reinforcing EIA practice in Canada [23]. EIA legislation has been developed in provinces and territories [24]. However, it has been shown that health is not adequately taken into account in EIA, limited to the analysis of potential health impacts of the determinants of the physical environment [25]. Health Canada published the four-volume Canadian Handbook on HIA which examines ways of integrating human health issues into the environmental assessment process [26]. In 2005, the National Collaborating Centre for Healthy Public Policy (NCCHPP) was established by the Government of Canada. The mission of the NCCHPP is to increase the expertise of public health actors across Canada in terms of healthy public policy through the development, sharing and use of knowledge. One of its mandates is to equip Canadian public health professionals for the prospective assessment of the potential impacts of public policies on the social determinants of health. The NCCHPP provides resources and training in HIA, develops tools to support HIA practice across the country and documents and disseminates practice-relevant examples [27,28]. In 2019, the entry into force of the Impact Assessment Act and the replacement of the Canadian Environmental Assessment Agency by the Impact Assessment Agency of Canada has strengthened the role of human health within impact assessments required by the federal government [29].

In Canada, the public health system is under the responsibility of the provincial governments. In British Columbia, Section 61 of the Public Health Act states that the Minister of Health “must evaluate, and advise the government on, those actions of government that may impact public health” [30]. In Quebec, Section 54 of the Public Health Act provides a framework for institutionalizing HIA at the central level of the government. This section reflects the government’s willingness to assess the potential impact of proposed legislation or regulations on the health of the population. In Ontario, the Ministry of Health and Long-Term Care has promoted the use of health equity impact assessment (HEIA) for identifying the potential health equity impacts of decision-making. The province of Nova Scotia has experimented with the practice of community HIA in a range of settings in the context of the People Assessing Their Health (PATH) Network [24,27,28].

Work on HIA has also been undertaken in some cities. The city of Toronto has conducted HIA on different topics such as municipal solid waste, island airport expansion and wastewater treatment plants, in partnership with Toronto Public Health [27,28]. The city of Saskatoon used HEIA to assess its development plan, in partnership with the Saskatoon Health Region and a non-profit organization called Upstream [31].

### 1.4. HIA in the Province of Quebec

In this province, Section 54 of the Public Health Act offered an opportunity to institutionalize public health functions in an innovative way which introduced a general mechanism to facilitate the consideration of health issues between ministries at the central level of governance [28,32], and thus foster the development of intersectoral policies. The Quebec Public Health Program for 2015–2025 called for the use of HIA during the review of land-use planning and development plans, in particular. In addition, the Quebec Government Policy of Prevention in Health (2016) explicitly refers to HIA as a tool allowing municipalities to integrate the assessment of potential health impacts in land-use planning. Although this is a favorable context to promote the creation of healthy environments, Quebec’s municipalities still have limited responsibilities regarding the population’s health. Public health and urban planning sectors are structurally set in silos to fulfil their own responsibilities and rarely work directly on the same interventions.

Before the development of the Government Policy of Prevention in Health, only two of the 18 health regions in the province of Quebec have used HIA [27]. Starting in 2011, under the leadership of the Regional Public Health Authority (RPHA) of Monteregie, a public health professional was responsible for finding projects for which HIA may be beneficial. His role, as a knowledge broker, allowed the creation of new collaborations between local and regional stakeholders, and the support of municipalities through a knowledge exchange process [33]. Nine of the HIA that were performed in Monteregie were evaluated. Although no consensus exists to identify what is a successful HIA [34], these HIA were positively appreciated by stakeholders [35]. However, no municipality has experienced HIA more than once. Quebec City is the second region involved in implementing HIA.

However, since 2018 and the implementation of the Measure 2.6 of the Government Policy of Prevention in Health, several other health regions are engaged in HIA, particularly in the municipal context. The Measure 2.6 aims to “Equip the municipal sector to more systematically integrate the analysis of potential health effects into land use planning and development processes” [36]. This is a way to encourage HIA practice at the local and regional levels in Quebec. In this context, several HIAs are currently being conducted by the RPHAs in partnership with municipalities and with the support of the Quebec’s Ministry of Health and Social Services and the Quebec’s National Public Health Institute [27].

### 1.5. HIA in Urban Planning

Land use and urban planning have important effects on the health and safety of populations [11]. The living environment, transport, accessibility, buildings, green spaces, local equipment, etc., have a direct or indirect influence on physical health (respiratory, cardiovascular pathologies, trauma), mental health (stress, depression, isolation) and health inequalities (social, environmental and territorial). The idea that urban planning and health are linked is not new [37]. Today, land use and urban planning projects too often focus on technical and environmental aspects and do not sufficiently consider health issues [38]. In addition, several effects of urban planning decisions on the population’s health are often ignored in the current practice of urban planning, although there is considerable interest in specific aspects of health, such as road safety [39]. In the context of an ageing population, the growing burden of chronic diseases (cancer, diabetes, cardiovascular disease, chronic respiratory disease), and the global epidemic of obesity, the use of the care system alone to solve these problems is insufficient. The complex relationships underlying population health, the production of health inequalities and urban planning call for a holistic approach that requires all sectors, health, education, planning, agriculture, economics and others, to collaborate in order to reduce the risks associated with these diseases and promote interventions that lead to healthy lifestyles and healthy social and environmental conditions. Research today is interested in developing mechanisms to strengthen the elements specific to urban health and healthy urban planning [40]. The deployment of the HIA approach is part of this. This tool supporting decision-making is a useful lever to take concrete action on the environmental, economic and social determinants of health in urban planning projects. HIA has been identified as a promising means of integrating health concerns into planning processes [41,42,43,44]. Urban planning is a good topic for the implementation of the HIA approach. In the US, as of January 2016, out of 386 HIAs carried out or in progress, about 70% were applied to decisions related to the built environment [45]. In New Zealand, out of 47 HIAs completed from 2005 to 2011, urban planning came first (about 28%) [46]. In Switzerland, out of 23 HIAs completed between 2001 and 2014, 26% were applied to decisions on urban planning [47]. In France, most of the HIAs are performed on urban development projects [44].

### 1.6. Objectives

The purpose of this article is to describe the approach developed for implementing HIA in the Quebec City’s urban planning process. The objective also is to draw lessons from this implementation to improve the integration of HIA into the urban planning process. Even though Quebec’s context has been presented above to illustrate in which conditions the HIA practice described in this paper were performed, the experience described could inspire others creating similar initiatives. Ultimately, the aim is to create a strong intersectoral collaboration to the benefit of population health and health equity.

## 2. Methods Undertaken to Create the Partnership

This paper provides the description of the approach used prior to a research project, presenting the steps for the elaboration of a partnership between organizations to perform HIA in Quebec City, based on direct observation by the researchers taking part in the meetings, and on meeting minutes. The minutes were written by the analytical team during the meetings. In 2013, through an academic initiative, a board of consultants was created to follow and to advise on urban planning projects proposed by the municipal representatives using the HIA methodology. The consultants were from the municipality of Quebec (QC), the Regional Public Health Authority (RPHA), the non-governmental organism (NGO) Vivre en Ville (VEV), and the National Collaborating Centre for Healthy Public Policy (NCCHPP). The consultants were initially consulted individually and have met twice a year as a group since 2015. Their responsibilities were to select urban projects for which HIA was judged relevant. Subcommittees were created for each project to identify health determinants that should be analyzed, to share expertise and concerns, and to contribute to the formulation of concrete recommendations to improve the project regarding public health-related issues. Using minutes from both committees’ meetings, we describe the perceived role of each institution before using HIA for the first time, how the HIA process was implemented, if it was appreciated by committee members, changes in their perception, and finally what outcomes were observed by the academic team.

No a priori strategy was set to implement HIA in the existing urban planning process of the municipality. Since pathways linking urban environments to health are particularly complex and the generalization from one setting to another can be problematic [4], an inductive approach was used in order to facilitate HIA implementation according to the local context, the political agenda, and the capacities of the involved stakeholders.

The following section (Results) presents the institutional arrangements and challenges the teams faced during the implementation of HIA, the approach that was developed to integrate HIA into an urban planning project in Quebec City, and the emerging characteristics perceived by participants in the HIA process that made it beneficial.

## 3. Results: The Partnership

### 3.1. Institutional Arrangements and Challenges for the Implementation of HIA

In 2012, Laval University’s Graduate School of Land Management and Urban Planning (GSLMUP) expanded its Master training program by including a new course aiming to teach public health concepts and tools to future urban planners. This professional training program needed a practical dimension linking urban planning and public health interventions.

The possibility of using HIA in a current project was explored with the municipality’s director of the Civil Security Department who was searching for new ways to improve citizens’ quality of life. Although assessing the potential impacts of a project on citizens’ health is not among the municipality’s responsibilities, the Civil Security director recognized one aspect in the definition of health that was linked to her mandate. Although health is among the most important priorities of the population, it was not presented as a final achievement or an ultimate goal, but rather as a resource that all citizens need for the development of a good quality of life. Thus, improving citizens’ health was not considered here as the mere reduction of diseases and trauma, in the same way as improving citizens’ security was not the mere addition of more policemen. It was in this analogous vision of health and security where the initial common ground rested between the researcher and the municipal administrator, when the first initiative of HIA was planned.

The first identified project was a green neighborhood project. Since this aimed to create a new neighborhood with respect to sustainable development principles, a positive evaluation of health determinants was expected. This was a deliberate choice to reduce any political concerns, while allowing for the first HIA experience. The Quebec City Urban Planning Department and the Regional Public Health Authority were first invited to collaborate. The agreement between the city, the university and the public health agency comprised three main conditions: (1) all information related to analyses (data, planning documents, etc.) will be kept confidential; (2) the city’s delegate will receive results from the analyses 30 days prior to the dissemination of the results; (3) the research team will retain complete intellectual property of the results.

The HIA green neighborhood project was performed by an inexperienced group of collaborators and a Master’s student in urban planning. The research team learnt from the existing literature on HIA, along with case studies, and was oriented by a professional from the NCCHPP. The lack of collective experience restrained the effectiveness of the HIA initiative and made the project lag for over two years. During this period, three Masters’ students and research professionals were involved, but none of these efforts allowed the production of substantial research results or useful recommendations for stakeholders. During the same period, the project was updated and delayed. Along with the involved stakeholders, the research team decided to continue the experiment, but using a different approach.

During the academic years 2014–2015, a group of four students from the new GSLMUP public health class decided to achieve their terminal Master’s project by completing an HIA on the same green neighborhood project. A larger group of stakeholders was solicited, including several professionals from the local community health services. The analyses took eight months to complete and an HIA report was presented to the stakeholders. The report was positively received, most recommendations were seen as appropriate and stakeholders from all institutions unanimously recognized their usefulness.

This positive experience convinced the municipality’s director of the Civil Security Department to support other HIAs for urban planning projects during an additional two-year period. This political and financial support strengthened the informal intersectoral collaboration that was initiated: it attracted the contribution of VEV, directly involved the NCCHPP, and provided the conditions for the addition of a postdoctoral researcher.

### 3.2. The Approach Developed to Integrate HIA into Urban Planning in Quebec City

Figure 2 shows the approach developed to incorporate HIA into the urban planning process in Quebec City. This first interinstitutional experience using HIA resulted in the creation of an informal group of stakeholders: the Quebec City Steering Committee on HIA (SC-HIA). The SC-HIA was composed of six members who decided which projects were to be conducted. These members were from the Quebec City Administration (QC), the RPHA, VeV, and the NCCHPP (Figure 2). These members had the responsibility of choosing from a list of 10 projects proposed by QC, and to delegate a competent professional from their organization to contribute to the working group according to the chosen projects’ characteristics. The SC-HIA was also composed of an analytical team of Laval University (LU) academics and Développement Santé (DS) (created by the students who performed the HIA on the green neighborhood project described above), a group of private consultants performing HIAs. While the analytical team was set up to coordinate the SC-HIA and the working groups’ activities, and to complete all HIA steps, they had no decisional power when it came to identifying projects, in order to keep an independent role in the process. The submitted projects were chosen by the SC-HIA and prioritized according to the projects’ timeframe and their institutional interests. Projects judged pertinent for an HIA were chosen through consensus by the SC-HIA members who decide which projects are conducted. The working group’s tasks were to provide detailed information on the project, help the analytical team gather the necessary data, describe useful information on the physical, social, cultural and economic local context, complete the analysis results, and validate the pertinence of the formulated recommendations. Depending on the project’s time frame, this contribution usually required two or three meetings. When the project under investigation involved a private promoter, such as in a housing development project, a representative was invited to take part in the working group’s activities.

After seven years, the SC-HIA led 10 HIA that were completed at two different scales (neighborhood, and street block level) (Table 1). All urban projects received clear, feasible and adapted recommendations. Some health determinants were assessed in all HIA: air quality, social capital, accessibility to green spaces, street walkability, and housing.

#### Data Collection and Appraisal

Regarding data collection, two types of data collection methods are used: (1) Local data analysis (tools: descriptive statistics, models, Geographic Information System (GIS); data: reports of citizens’ consultations, information provided by the municipality (i.e., infrastructure and land use documentation of the project)); and (2) Scientific knowledge (tools: literature review on the health effects of urban elements on determinants of health; data: consultation of similar HIA reports). The appraisal methodology usually follows two steps: (1) evaluate the urban elements of the projects that could impact health, and (2) evaluate the type and intensity of health impacts (see Step 3. Appraisal in Figure 1).

An HIA report was then produced by the Analytical Team and bonified with the input of the working group. Results were then presented to the SC-HIA. All reports comprise five main parts: (1) a summary of the project, (2) a description of the initial situation, (3) a description of the potential effects of the planned interventions on health determinants (e.g., street walkability on physical activity), (4) a synthesis of the scientific literature on the known impact of these determinants on human health (e.g., physical activity level on cardiovascular risk), (5) a list of specific recommendations for each of the project’s objectives. Recommendations not only targeted threats to health, but also positive interventions that could be further enhanced at a low cost. Table 2 shows examples of recommendations and how they were applied by the municipality.

### 3.3. Emerging Characteristics Perceived by Participants in the HIA Process

Stakeholders (involved the working group committees) from all institutions provided feedback following their involvement in the HIA process.

QC administrators initially accepted the implementation of HIA in a pilot experiment for two reasons: (1) it was proposed by an independent academic who had no political constraint and could work according to confidentiality engagements; (2) health was not the ultimate purpose of the project, but was rather considered as an essential resource to improve the citizens’ quality of life, which is part of the municipality’s field of action.

Urban planners more directly involved in the HIA process reported that it brought abundant new and relevant information on the impact of their work on public health issues and was perceived as very helpful in fulfilling their responsibilities. They further used the information from the scientific literature and the field observations provided by the analytical team to convince other urban planners within their own department to change several choices that had usually been made without a clear justification. Some of them also proposed that HIA could be introduced slightly earlier in the urban planning process to use HIA results in public consultations.

The RPHA stakeholders and the community workers particularly appreciated working with urban planners on a positive project. While many of these initially saw the HIA as a powerful advocacy tool to enforce public health interests, those who participated recognized that collaborating more directly with urban planners allowed for a better understanding of their responsibilities and concerns. Some reported that the regular intersectoral collaborations contributed to the development of a common language between institutions, which facilitated their cooperation on other projects. Their first HIA experiences greatly helped consolidate their own responsibility in identifying and evaluating potential risks to the health of the population in urbanistic projects. They also mentioned a clear intent to develop their own expertise in HIA to intervene on other subjects apart from urban planning (e.g., social policies), while keeping their involvement in the SC-HIA.

VeV professionals’ official mandate is oriented toward sustainable development. Sometimes, the projects they would have proposed were different from those submitted to HIA. Yet, they recognized that the main principles of sustainable development are among the main social determinants of health. Their involvement in several HIA processes has shown them a potential tool for the promotion of sustainable initiatives and they are also considering the development of their own expertise regarding HIA.

The NCCHPP representative greatly appreciated this initiative since it directly contributed to the institution’s mandate. The involvement of an interdisciplinary group of Master’s students was highly valued, not only because it helped bridge public health interests and urban planning in the completion of current projects, but also because it contributed to the preparation of the next generation of urban planners to work at the interface of both institutions.

Two private promoters were involved in a working group. Neither of them had ever heard of HIA before and showed curiosity, if not suspicion, regarding the fact that their project would be scrutinized by academics. Both promoters, however, recognized very quickly that HIA could improve their project at a low cost and facilitate social acceptation of their project. Recommendations were highly appreciated and led to many positive modifications improving the original project. One of the promoters directly used the HIA report as a negotiation tool with a provincial ministry to facilitate the implementation of the project and to show respect for environmental norms.

The last SC-HIA meeting was held in May 2020. The members who decided which projects would be used for HIA identified three more projects for which they proposed to perform an HIA in the coming year. They further recommended taking the time to perform a systematic evaluation of all HIAs that were actioned in the region and to collect information on the use of recommendations and on the appreciation of the process.

## 4. Discussion

Implementation of HIA in the urban planning process came from the complementary needs of a practical dimension, linking urban planning and public health interventions for a Master’s training program and the municipality that was searching for new ways to improve citizens’ quality of life. The objective of this paper was to describe the approach developed for implementing HIA in the Quebec City’s urban planning process and to draw lessons from this implementation to improve the integration of HIA into an urban planning process.

For the ten projects, implementing HIA in the urban planning process was perceived as a positive and useful tool for decision-making. Participants reported that the regular intersectoral collaborations brought out many expected and unexpected outcomes as a direct result of adding the HIA process.

Although no a priori strategy was set to implement HIA, one could have expected outcomes somewhat implicit to HIAs, such as facilitating knowledge transfer and data sharing between institutions, developing stronger arguments for decision-making among urban planners, and highlighting potential social inequalities

This first experience in using HIA in the urban planning process also brought out unexpected outcomes such as the creation of a private corporation of consultants on HIA (DS), private promoters using an HIA report to negotiate with other institutions, institutions studying the possibility of developing their own expertise in HIA, and the possibility of presenting HIA analysis results in public consultations.

The HIA process may also have contributed to the development of a regional intersectoral community of practice. As public health and municipal institutions’ mandates are structurally divided into independent functions, the HIA process allowed stakeholders to share information and concerns regarding their responsibilities. The fact that these stakeholders voluntarily accepted participation in many projects suggests that the HIA process answers to a current need. Although HIA is an additional step in an already complex procedure, we observed that none of the stakeholders perceived HIA as invasive or cumbersome. We also witnessed that performing HIA analyses took less time to complete over time, with increased experience of the SC-HIA committee members.

Up to now, the research team judges the approach developed for implementing HIA in the Quebec City’s urban planning process as beneficial for all involved parties. This rather positive intersectoral collaboration was possible for several reasons. First, performing HIA was possible because the research team was politically supported by the municipality’s stakeholders who saw HIAs as a possible solution to complex problems in urban planning interventions. Second, health was not presented as an ultimate goal, but rather as a resource that is essential to the citizens’ quality of life. This vision of health fosters intersectoral collaboration by putting the responsibilities of each institution on more equal grounds. Third, integrating HIA was proposed by academics who have no responsibility regarding public health or urban planning decisions. Having independent coordination in the process has possibly facilitated intersectoral collaborations. Fourth, adding HIA as a voluntary procedure implies an interest from participants in the collaboration. Imposing a mandatory procedure might not bring the same stakeholders around the table and negotiations could have taken a different turn [22,32]. Fifth, HIAs were conducted in a flexible way that adapted to the project’s timeframe without slowing down the decision-making process, which reassured the involved actors. Sixth, the HIA initiative was perceived as answering a variety of needs that are not necessarily the same for each institution (mutualism). This observation is in line with the Ottawa Charter’s principles, the Health in All Policies approach [48]. It is further supported by the new Quebec Government Policy on Health Prevention (2016) which explicitly refers to HIA as a tool allowing municipalities to better integrate the assessment of potential health impacts in land-use planning processes.

This policy is partly aimed at extending the Quebec government’s pursuit of Health in All Policies [49] down to the level of regional and municipal governments. Applying HIAs with a structured community of stakeholders appears to have left a legacy of awareness and capacity for working intersectorally that could facilitate new practices and opportunities for collaboration locally. For example, here and in other localities with such prior experiences of using HIA, (e.g., Monteregie), we might expect that the cross-sectoral understanding of the bidirectional relationship between population well-being and the interests of other sectors [50], and the trusting and effective cross-sectoral working relationships, might facilitate the identification of “win-win” arrangements that link partners from different sectors like urban planning and health [51].

The lack of experience and the skepticism of private promoters at the beginning of the process were stated challenges. We also need to acknowledge the difficulty related to the community’s engagement. When short HIA take place, it is even more challenging to make sure that the population participates in the HIA process due to lack of time. Fortunately, in our HIA working group we could count on the help of a community worker whose expertise allowed a better understanding of the reality and needs of local populations. When available, the results of public consultations planned by the municipality were also used.

Here are some lessons drawn from the Quebec City experience in integrating HIA into an urban planning process:Promote the conduct of the HIA process in conjunction with an environmental assessment (EA) when the latter is required for the proposal under review. This will allow HIA to benefit from the experience of EA in urban planning processes and will provide an opportunity to exchange data necessary for carrying out both impact assessments.Formulate, wherever possible, recommendations from the HIA directly on the proposed draft urban plan.Develop a plan or a strategy to follow up on the effectiveness of recommendations made during the HIA process. This helps to develop the planners’ understanding of the actions proposed to improve their project in terms of health and to facilitate the consideration of these actions.Having a neutral/independent actor with some credibility, such as a university, to coordinate the HIA process may help facilitate intersectoral work and seems to facilitate the acceptance of HIA results by all the actors involved. With a neutral actor, there are no conflicts of interest.For the actors in charge of conducting HIAs in urban planning, make sure that everyone knows how to work on urban plans and maps to facilitate interactions with planners.Have some flexibility in integrating local community’s concerns into the HIA when the time to complete the impact assessment is very short (for example, three months). This could be considered as a rapid HIA. In this type of HIA, it is often difficult to keep some time open to enable the population affected by the proposal under review to participate in the HIA process. One solution would be to include in the HIA working group a person with an expertise of the local communities’ needs and territory. Another way would be to use the results of public consultations organized by the city on the project under review, when these are available.

These lessons from Quebec’s experience show how the implementation of HIA projects can address the particularly complex challenge of integrating public health issues in urban planning. This experience allowed us to develop a strong research partnership with a variety of actors to perform HIA projects as well as to evaluate the results. Further research will allow us to improve our process as well as the scope of the performed HIA projects.

## 5. Limitations

This paper does not portray a study or HIA per se to evaluate the process. It shares the steps undertaken to create a shared space where actors concerned with urban planning can use HIA to improve urbanistic projects. The evaluation of the performed HIA is ongoing. The authors’ goal was the share Quebec’s experience for it to inspire others who would like to develop a similar experience; the approach used to formally perform HIA with many organizations is promising.

## 6. Conclusions

Although some difficulties were faced in the first years, the introduction of HIA in the urban planning process in Quebec City proved to answer appropriately to emerging needs. It induced a variety of expected and unexpected outcomes from which emerged an intersectoral community of practice aiming to improve the quality of life of citizens. Although the SC-HIA (Quebec City Steering Committee on HIA) remains an informal group, the sum of the observations provided in this article suggest that the stakeholders involved in HIA in the Quebec City region are currently changing practice based on the development of a collective expertise. This experience supports the idea that academics may have an important role in supporting governmental institutions, NGOs and private corporations at the regional level to facilitate intersectorality, mutualism and, ultimately, foster collective performance. HIA is not the only way to combine public health and urban planning interventions, but it appears to be relevant in the Quebec City context. Since other HIA projects have been requested by the SC-HIA, it would be interesting to evaluate the Quebec region experience more thoroughly. The positive results generated by the HIAs carried out in Quebec City and the new Regional Public Health Action Plan (2016–2020) offer new perspectives for the elaboration of an intersectoral partnership to perform health impact assessment in urban planning, and provide a better understanding on how Health in All Policies (HiAP) could be implemented at the local and regional levels in other Canadian contexts.

## Figures and Tables

**Figure 1 ijerph-17-07556-f001:**
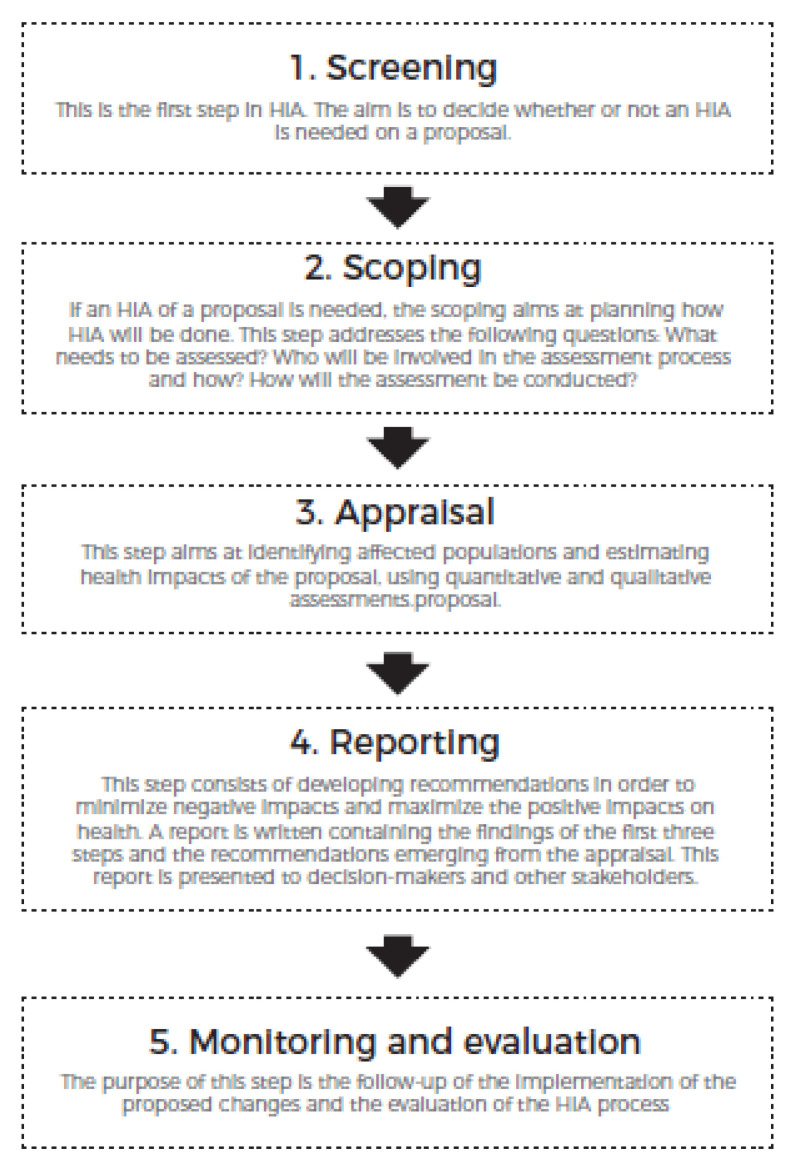
The five steps of the health impact assessment procedure.

**Figure 2 ijerph-17-07556-f002:**
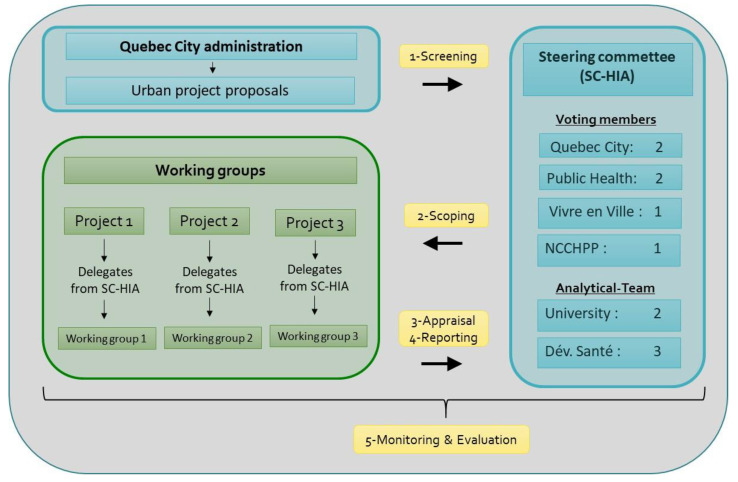
Approach for Health Impact Assessment (HIA) implementation into the urban planning process in Quebec City.

**Table 1 ijerph-17-07556-t001:** Characteristics of the health impact assessments performed between 2013 and 2020 in the Quebec City Region.

HIA Projects		Sectoral Representatives								HIA Report		
HIA Project	Scale	QC	RPHA	VEV	NCCHPP	Promotor	DS	University	Student	Start	Completed	# Recommendations
Green Neighborhood	site	2	2	0	1	0	0	1	9	May 2013	May 2015	22
Local urban plan 1	local	3	6	0	1	0	0	2	5	Sept. 2015	May 2016	24
Housing project	site	3	2	2	1	1	3	2	0	April 2017	June 2017	24
Local urban plan 2	local	3	3	2	1	0	3	2	1	June 2016	Jan. 2017	42
Housing project	site	3	3	2	1	1	3	2	0	Sept. 2017	Jan. 2018	41
Industrial and high technology development plan	site	3	3	1	0	0	3	2	0	March 2018	Sept. 2018	39
Local street	local	3	2	1	0	0	4	2	0	July 2018	Dec. 2018	23
Public transport	site	1	4	1	0	2	0	2	4	Sept. 2018	May 2019	59
Public space	local	2	1	1	0	0	3	2	1	Jan 2019	May 2019	21
Park	local	3	2	1	0	0	3	2	0	Sept. 2019	April 2020	21

**Table 2 ijerph-17-07556-t002:** Examples of recommendations provided in the completed HIAs and their application by the municipality.

Recommendation	Application
Separate the dog park in two with an opaque barrier, one part for small dogs and the other for big dogs, which should discourage barking.	The municipality used this recommendation and all related explanations when residents asked which improvements would be made to reduce the noise level related to the dog park.
Universal accessibility: favor mobility using a wheelchair and improve the comfort of pedestrian infrastructure by installing 2.4m-sidewalks on each side of the road.	The concept was presented to a committee. Modifications have thus been applied to allow for living spaces for vulnerable populations in the project.
Pedestrian safety: adjust the speed limit according to the ambient characteristics to be coherent with the environment (30 km/h for the residential neighborhood).	This was a major preoccupation for the municipality. This recommendation allowed for an insistence on the application of measures for the safety of pedestrians. All parties involved in the project were asked to consider pedestrians.
Lighting: there should be 5 m between lamp posts. Blue/white light should be diminished, and yellow light should be favored to reduce the impact on the circadian rhythm of the residents. 6 lux for pedestrian and cyclists’ areas and 8 lux for residential streets.	This allowed the reconsideration of norms regarding lighting in residential zones for better lighting and not more lighting.
Favor urban agriculture: create gardens on the roofs to favor access to fresh fruits and vegetables for disadvantaged populations.	Community gardens were integrated in the project since the HIA provided a space to debate this with stakeholders, promoters and municipal officials.
Social diversity: ensure a certain level of social diversity by adding nine social housing units to represent 10% of the total number of units.	It was decided that within the 700 units that would be developed, 70 would be for social housing (10%) which was recommended in the HIA.
Air quality: increase the green area of the neighborhood to increase the air filtration capacity of the neighborhood.	Greening and adding to green areas have been considered to reduce the bad air quality.

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
