# Peer review of "The Elaboration of an Intersectoral Partnership to Perform Health Impact Assessment in Urban Planning: The Experience of Quebec City (Canada)"

_ijerph, 2020, doi:10.3390/ijerph17207556_

Round 1

Reviewer 1 Report

The authors have addressed the issues i had raised as per the attached 

Reviewer 2 Report

I consider that the revised version of the article entitled: "The elaboration of an intersectoral partnership to perform health impact assessment in urban planning: the experience of Quebec City (Canada) ", generally took into account the comments of the reviewers and improved the way of understanding and presentation of the results of the article.
The esteemed authors enriched their article with further bibliographic references that document those points that needed revision based on the above remarks.
The above reasons but also the way the authors managed the intermediate time, from the first revision until today, make me suggest its publication in the present form.

Best regards,

Reviewer 3 Report

I would like to sincerely acknowledge the excellent revision as provided by the author's. This is now a piece of work that is suitable for an international audience and is of great value to the practice of HIA. Excellent work. A pleasure to read. 

This manuscript is a resubmission of an earlier submission. The following is a list of the peer review reports and author responses from that submission.

Round 1

Reviewer 1 Report

Please find  the review comments for improvement of your article:

The article : Using health impact assessment in the urban  planning process in Quebec City (Canada): outcomes  and implications after five years

Abstract: Health impact assessments (HIA) allow evaluation of urban interventions’ potential  effects on health and facilitate decision-making in the urban planning process. However, few  municipalities have implemented this method in Canada. This descriptive case study reports the  process and outcomes of HIA implementation after five years of interinstitutional collaborations in  Quebec City (five HIA). Using direct observation and meeting minutes, this information was  described: perceived role of each institution taking part in HIA before using it, how the HIA process  was implemented, if it was appreciated, and which outcomes were observed. The intersectoral  interactions contributed to the development of a common language, which sped up the HIA process  over time and fostered positive collaborations in unrelated projects. It was an effective tool to share  concerns and responsibilities among independent institutions. This experience resulted in the  creation of an informal group of stakeholders from four different institutions that keep performing  HIA

The role of Health Impact Assessment (HIA) as a tool in evaluation of policy and programmes in all sectors  and enhanced recognition of societal determinants of health  has grown over the 4 the past four decades  hence your study “Using health impact assessment in the urban  planning process in Quebec City” has potential to advance knowledge on the tool especially in urban planning processes. However You need to state the research design ( which is evaluation design ) and adhere to  the basic framework on how evaluation studies are presented i.e.

context including situation analysis of HIA in Quebec vis avis the use of other planning tools especially environmental assessment ( EA) in urban planning the project of your concern conceptual framework/evaluation framework adopted Risk governance and/or regulatory framework on urban planning

Here are some suggested articles:

Briggs , D.J( 2008).A framework for Integrated  environmental health impact assessment of systemic risks. Environmental Health

Hanna  and Cousens, 2001

I make a few observations and which can be used in the improvement of the article as given hereunder:

Abstract : Missing section on implication of your study and research design used

Title :   Will the title change if you removed…….. after 5 years

Merge line 70-78 under 44-52 as it is repetition

You need to provide one line synthesis of the proceeding section after line 86, otherwise it is hanging)

Line 88- You need to specify what is action research ( in your own words)

Line 91-92   becomes hazy when you say “our short report”. Is this a study or a report?

Line 117-127 cannot be  a result/ finding

Figure 2  is not clear and source missing

Line 195-198: Figure is not clear and source missing

Line 204  just states the number of recommendations  but does not provide details. You need to  summarise  the given recommendations into themes across board

Line 290-292: The source of the statement not provided

Line 321- 342: What informs this  yet it is not part of the objectives and background?

Line 322: How is this different  from previous studies  on HIA? Are EAs  not  part of urban planning in Quebec?

Line 366 State how this comes about

Line 338 Seems to be contradictory…. Doesn’t HIA include community participation?

Specific comments

1.Organisation

There is need for reorganization of the article to reflect the nature of the study( evaluation design and its components missing) and to have a coherent flow from one section to another  

2.Are sections well developed;

The sections are well developed but missing key points for it to flow ( refer my general comments in i- iv above)   

Is the literature well synthesized

There are gaps on  literature concerning the use of the term evaluation ( what type of evaluation i.e. diagnostic, prognostic or summative). This mainly arise largely due to failure by the authors to operationalize the  term evaluation  and its various dimensions. You may refer Bhatia and Wernham. Integrating Human Health into Environmental Assessments:  An  unrealized  opportunity for  Environmental Health and Justice 

Does the author answer the questions he/she sought to answer

The authors have made a good attempt at  answering the questions he/she sought to answer but they can improve the manuscript by first operationalising the term evaluation and its context as referred i-iv above 

Is the method well explained:

The methods must  be improved by providing a conceptual framework on evaluation study used and its context ( see i-iv above ).

The methods are inadequately described i.e. what type of evaluation study, variables  and their interaction and assumptions

Data sources are inadequately described

Is the article well written and well understood

Except for the gaps on conceptual framework , the article is well written  and can be understood. However there is need to reorganize the article to reflect all the comments from 1-5 if coherence is to be achieved

Concluding remarks

Evaluation studies are taken within a given context which should be well described. This includes the framing and key variables used , their relationship and assumptions taken. The intervening , independent and dependent variables ( flow diagram) should thus be stated clearly. The limitations  and challenges should also be stated. In turn, this guides  the synthesis and integration of the findings. The article has attempted to do the above but in disjointed  manner.  I thus recommend that the authors relook at the organisation and conceptual framework and rearrange the article.   The conclusion need to be shortened by restating the objective(s) and providing the main findings.

Kind regards

Reviewer 2 Report

I would classify the article to be reviewed as information - bibliographical article on the implementation of HIA in order to assess the potential impacts of urban health interventions and facilitate decision - making in the planning process.
The honorable authors describe the processes, failures, and results of the method by referring to its application in Quebec City, Canada, making the best use of both existing bibliography and reports from its application over time.
Although the article shows a prototype stemming from an indirect objective of urban planning (ie the good health of citizens resulting from its correctness and completeness), however, the extensive description of my intermediate procedures created many ambiguities, especially as to how the method is ultimately able to affect the health of the population. I could not really understand, either qualitatively or quantitatively, how health was considered an indispensable resource for improving the quality of life of citizens.
In the end, I think that the descriptive (and I would say technocratic) approach of the article did not work.
Although the HIA's methodology itself provides a useful tool for planners to promote public health when designing a city or a part of the city, the qualitative and quantitative dimensions of the results of its implementation should be tangible and understandable.
For this reason, I do not agree with the publication of the article.

Reviewer 3 Report

Thank you for the opportunity to review this manuscript on HIA. I have found this to be an interesting case however, given the international audience associated with the journal, the author's should be prepared to present a more solid product as to how this case fits into the global application of HIA in Urban Planning. At present, the way it reads, is as a regional report, lacking the appropriate global references and failing to comment on what this work and experience holds for the global urban planning community. 

Introduction:

HIA while perhaps new to Quebec, has been applied to urban projects prolifically in Europe and Asia. This should come out clearly in the introduction as well in the discussion. Canada is 10 years behind the times and this deficiency needs to come out. 

The publication is missing some key Canadian HIA references including the 2004 Canadian HIA Guideline publication and the MetroVancouver HIA Guideline.

As aforementioned, there has also been a lot of work on health and HIA guidelines in the urban planning context. See early WHO 1999 piece on urban health, 2018 ADB HIA framework for economic zones (industrial urban areas) as well as the massive recent Tsinghua-Lancet commission report for the PRC. An example citation is provided. 

L.J Duhl and AK Sanchez. 1999. Healthy cities and the city planning process: A background document on links between health and urban planning. World Health Organization, Copenhagen, Denmark. Available online at: http://www.euro.who.int/__data/assets/pdf_file/0009/101610/E67843.pdf

The intro should draw early linkages to urban planning and health and a comment as to why urban planning failed to keep considering health as a central part of the mandate would be an interesting detail for the reader.

Another important piece of literature is missing that the author's should consider incorporating:

Bird, E., Ige, J., Burgess-Allen, J., Pinto, A. and Pilkington, P. and Public Health and Wellbeing Research Group. 2017. Healthy people healthy places evidence tool: Evidence and practical linkage for design, planning and health. Technical Report. University of the West of England, Bristol. [In Press] Available online at: http://eprints.uwe.ac.uk/31390

Methods/Results:

It would be appreciated to understand how the HIA was actually conducted (implemented). The data collection and analysis piece in this type of arrangement is perhaps a very interesting aspect of the case from a scientific perspective and it would be good for the authors to describe this in more detail. For instance section 3.2's title is HIA Implementation - rather the content describes institutional arrangements and challenges. Was an independent ethics board involved? What were the main scoping issues that emerged? What kind of baseline data was collected-used-analyzed? These are major challenges for HIA practitioners globally and would make the manuscript a more relevant and meaningful read. 

Discussion/Conclusion:

Again, given the importance of this topic globally and across Canada, and the international reach of the journal, I recommend the authors' extend this case and place it in the broader Canadian and global context and clearly articulate what it has contributed to HIA practice in Urban planning.